# Turfgrass Salinity Stress and Tolerance—A Review

**DOI:** 10.3390/plants12040925

**Published:** 2023-02-17

**Authors:** Haibo Liu, Jason L. Todd, Hong Luo

**Affiliations:** 1Department of Plant and Environmental Sciences, Clemson University, Clemson, SC 29634, USA; 2Department of Genetics and Biochemistry, Clemson University, Clemson, SC 29634, USA

**Keywords:** turfgrass, salinity stress, stress, turf soil, salinity tolerance, turfgrass drought stress, turfgrass stress tolerance, turfgrass roots, turf

## Abstract

Turfgrasses are ground cover plants with intensive fibrous roots to encounter different edaphic stresses. The major edaphic stressors of turfgrasses often include soil salinity, drought, flooding, acidity, soil compaction by heavy traffic, unbalanced soil nutrients, heavy metals, and soil pollutants, as well as many other unfavorable soil conditions. The stressors are the results of either naturally occurring soil limitations or anthropogenic activities. Under any of these stressful conditions, turfgrass quality will be reduced along with the loss of economic values and ability to perform its recreational and functional purposes. Amongst edaphic stresses, soil salinity is one of the major stressors as it is highly connected with drought and heat stresses of turfgrasses. Four major salinity sources are naturally occurring in soils: recycled water as the irrigation, regular fertilization, and air-borne saline particle depositions. Although there are only a few dozen grass species from the *Poaceae* family used as turfgrasses, these turfgrasses vary from salinity-intolerant to halophytes interspecifically and intraspecifically. Enhancement of turfgrass salinity tolerance has been a very active research and practical area as well in the past several decades. This review attempts to target new developments of turfgrasses in those soil salinity stresses mentioned above and provides insight for more promising turfgrasses in the future with improved salinity tolerances to meet future turfgrass requirements.

## 1. Introduction 

Turfgrasses are ground cover plants with intensive fibrous roots sensitive to different edaphic stresses. Urban green spaces are turfgrass-dominated and backgrounded landscapes of varying functions and uses that are ubiquitous in areas associated with human population growth and urbanization in the United States. Furthermore, lawns cover 1.9% (approximately 186,800 km^2^) of the country’s terrestrial areas [1,2], with about 80 million lawns sitting within tropical to subarctic climatic zones. Among the mentioned edaphic stresses, soil salinity is one of the major stressors, and it is highly connected with drought and heat stresses of turfgrasses as well as other stresses [3,4] (Figure 1). This review is focusing on the recent development in turfgrass salinity stress and tolerance literature updates and facing the challenges of shortage of turfgrass irrigation water and climate changes.

Turfgrass is perhaps one of the most important vegetative ground covers in the world as it provides functional, recreational, and ornamental purposes to our landscapes, particularly in urban communities. Approximately 50% of global land area is categorized into one of the four classes of aridity [5], and as the global effects of climate change continue to intensify, the demand for freshwater has drastically increased in the recent years. Along with numerous government regulations restricting freshwater use for turfgrass and crops alike, many managers have turned to non-potable, secondary (recycled), and saline water sources. The presence of dissolved salts in non-potable water in addition to poor soil conditions can lead to saline or sodic soil conditions and reduced plant growth. Plants exhibit many morphological, physiological, and metabolic responses to salinity stress that can significantly decrease crop yield and quality [4,5]. Thus, salinity is one of the most significant soil stress conditions limiting plant growth and productivity all over the world [6].

Saline soils and salinity stress are often confused with other terminology such as alkaline and sodic soils, and it is important to understand the terminology before reading any literature on salt stress on plants because two terms could imply different meanings. Soils that have high concentrations of neutral soluble salts that are detrimental to plant growth are saline and capable of imposing salinity stress on plants. In contrast, alkalinity and alkaline soils strictly refers to the concentration of hydroxyl (OH^−^) ions and soils with a pH > 8.5, respectively (Table 1). Sodic soils are most commonly confused with saline soils, but the difference between the two is that sodic soils have high concentrations of sodium (Na^+^) and chloride (Cl^−^) salts (ionic soluble salts) that are negative and even detrimental to plant growth [7].

## 2. Causes and Factors Affecting Soil Salinity for Turf

Salts often occur naturally in soil at non-toxic levels, however there are many factors contributing to the cause of salinity stress in plants [8,9,10]. Soil salinity is caused by a variety of natural processes including rock weathering and fluctuating depth of the water table. Weathering dissolves soluble salts from the composition of rocks causing the solutes to move with water through the soil solution [7,11]. The fluctuating depth of the water table determines where the soluble salts will be in relation to the root zone, and therefore can directly impact plant growth and soil structure. In addition, many environmental, soil physical, and chemical properties all affect the capability of soil to accumulate salts and the susceptibility of turfgrass species to salinity stress. Some environmental factors affecting salinity tolerance include temperature, water status, light, soil depth, soil particle fineness, timing, and depth of irrigation [12]. Hot, dry conditions make turfgrasses more sensitive to saline conditions because of increased evapotranspiration rates favoring salt uptake [4]. Decreases in soil water content show an increase in root zone salinity due to the water potential gradient created by plant roots that pull water towards the root zone. Shallower depths of irrigation increase root zone salinity due to a lack of sufficient water to flush/leach salts out of the root zone. In addition, extended periods of time between irrigation events also increase soil salinity because the groundwater is pulled up the soil profile due to the reduced matric potential in the upper horizons. The rate at which salt accumulates in soil depends on the concentration of total dissolved salts in irrigation water, annual amount of rainfall and irrigation water, the soil’s physical and chemical properties, and efficiency of landscape drainage. Salt accumulation occurs more rapidly in soils with thick thatch layers and in clay-dominant soils that have low permeability and are prone to compaction [12].

The quality of irrigation water used is perhaps the most important factor when trying to control soil salinity and reduce plant salinity stress. As available water supplies continue to diminish, turf practitioners are looking for alternative irrigation water sources [13]. Many have turned to recycled and non-potable water sources to irrigate turf landscapes; however, these can have adverse effects on soil’s chemical and physical properties including soil salinity [12]. Recycled water, also referred to as recycled wastewater or effluent water, is water that has gone through one cycle of human use and is sent from a home or business through a sewer system to a sewage treatment plant. Recycled wastewater will have a higher concentration of suspended solids and various nutrients, thus resulting in varied soil parameters such as sodium adsorption ratio, exchangeable sodium percentage, and pH. All of these influence soil salinity and nutrient availability. There are no consistent concentrations of substances in recycled wastewater due to seasonal and annual variability of trace contaminants in the initial wastewater [14]. However, recycled wastewater generally contains greater concentrations of sodium, bicarbonates, carbonates, boron, and chloride, which can all significantly impact soil’s chemical properties and fertility [15]. Due to the differences in nutrient concentrations between recycled wastewater and potable water, soil infiltration rate and hydraulic conductivity are often negatively impacted. A 50% decrease in infiltration rate after three years of recycled wastewater irrigation was observed on golf course fairways due to interactions of dissolved salts and sodium in the recycled wastewater with soil particles [16]. Suspended solid deposits that are physically clogging soil pore spaces is a common cause of reduced soil porosity while under saline conditions of recycled wastewater irrigation. The use of effluent and non-potable water does have greater effects on horticultural crops than turfgrasses due to the lower amount of above-ground biomass that is more sensitive to salinity [17,18]. In addition, many turfgrass species and varieties (particularly C_4_ turfgrasses) are relatively tolerant to the salinity of recycled wastewater if applied directly to soil. However, excess foliar absorption of salts accumulates much quicker in leaf margins and shoot tips when compared to root absorption.

Salts are added in standard application of fertilizers and soil amendments, even including some pesticide applications [12,19]. Naturally, coastal areas that have evaporated and left salt deposits at the surface or in layers in the soil profile suffer more turfgrass salinity stresses than inland areas [19]. Although there is a lack of studies specifically focusing on turfgrasses, airborne saline particles are a heterogeneous group of chloride (Cl)-containing airborne materials both from natural as well as anthropogenic origins [20]. These are air pollution problems primarily relating to source and dispersion, but these airborne saline particles can cause the accumulation of Cl on a vegetation canopy, negatively affecting the growth and performance of turfgrasses (Figure 2).

## 3. Turfgrass Responses to Saline Stresses

Individual plant species will respond differently to salinity. Ornamental and vegetable plants are known to be more sensitive to saline conditions, while most grain crops and turfgrasses are considered more tolerant to salinity. Saline conditions trigger a wide variety of physical and chemical responses inside the plant. Most of the detrimental effects of salinity are due to symplastic osmotic pressure fluctuations causing cellular water loss and reduced water uptake. Additionally, the disruption of ion balances, various ion toxicities, and typical chlorotic leaf tissue creates adverse effects on plants metabolic processes. Therefore, plant growth and development are often limited or inhibited while under salinity stress [8,9,10].

Shoot and root growth responses are perhaps the most documented and recognizable plant responses to any environmental stress, especially soil salinity, in addition to yields as crop plants. A negative linear relationship between shoot growth and salinity is often reflected in salt-sensitive and moderately salt-tolerant species. Salt-tolerant species such as *Cynodon* spp., *Paspalum vaginatum*, *Puccinellia distans*, and *Zoysia matrella* can show stimulated shoot growth during exposure to moderately saline conditions [21,22,23]. Turfgrass quality and leaf width and length significantly decreased in four species of cool-season turfgrasses (*Lolium perenne* ‘Belida’, *Festuca rubra commutata* ‘Casanova’, *Poa pratensis* ‘Evora’, and *Festuca rubra trichophylla* ‘Smyrna’) when irrigated with gradually increased salinity levels from 0.54 mM to 200 mM NaCl [24]. The same observations were found in two centipedegrass (*Eremochloa ophiuroides*) accessions, E092 and E092-1, as salinity increased from 0 mM to 75 mM NaCl [25]. Seed germination is also affected by soil salinity, as four cool-season turfgrass species (perennial ryegrass, slender creeping red fescue, tall fescue, and Kentucky bluegrass) had an 11% to 25% decrease in field germination at the highest salinity level of 200 mM NaCl [26]. Certain management practices may also influence a species’ salinity tolerance, such as mowing height. Lower mowing heights of ‘L-93′ creeping bentgrass (*Agrostis stolonifera*) severely decreased salinity tolerance as determined by visual turf quality. When mowed at 6.4, 12.7, and 25.4 mm, the turfgrass reached an unacceptable visual quality at soil EC values of 4.1, 12.5, and 13.9 dS m^−1^, respectively [27]. These results suggest that a higher shoot biomass increases turfgrass salinity tolerance due to the presence of more plant tissue able to sequester saline ions in vacuoles. The same study found that as soil salinity levels increased, the treatments with the highest mowing height of 25.4 mm retained a higher percentage of original root biomass under saline conditions when compared to the lowest mowing height of 6.4 mm [27]. Most turfgrass species that are considered moderately tolerant to tolerant of salinity stress often demonstrate an increase in root/shoot ratio while under salinity stress [21]. This is due to the plant increasing or maintaining rooting, while shoot growth decreased in response to salinity and may act as a tolerance mechanism. All species (alkaligrass, saltgrass, tall fescue, and Kentucky bluegrass) exhibited an increase in root/shoot ratio at all salinity levels. Of the four species, only saltgrass (*Distichlis spicata*) demonstrated a relatively similar shoot mass as soil EC increased from 2.5 to 25 dS m^−1^. An initial increase in relative root mass was observed for both alkaligrass and saltgrass up to 15 dS m^−1^, while a maintenance of root mass was observed for tall fescue at the same EC [21]. The same stimulated root growth was observed in a halophytic grass species, *Sporobolus virginicus*, as NaCl concentrations increased up to 1.0 M [28].

Figure 3 summarizes the metabolic processes associated with salinity tolerances and sensitivities among turfgrass species and cultivars. However, the primary mechanism behind shoot and root growth responses to salt stress is due to the inhibition of photosynthesis. Salinity stress causes a reduction in CO_2_ assimilation to the chloroplasts due to stomatal closure. Guard cells are responsible for controlling stomatal aperture to regulate water and gas exchange. Due to the structure of guard cells, they respond to changes in turgor pressure, therefore stomates close because of decreased osmotic pressure and reduced K^+^ content in leaves [29]. Increased shoot concentrations of Na^+^ and Cl^−^ cause increased osmotic pressure, resulting in cellular water outflow or a lower turgor pressure. As a result, the osmotic potential inside the cell decreases [30]. Salt-sensitive plants often exhibit decreased chlorophyll content through inhibition of chlorophyll synthesis and enzyme-activated chlorophyll degradation resulting from oxidative stress [31]. Total chlorophyll concentrations of three cool-season turfgrass species (*Lolium perenne* ‘Belida’, *Poa pratensis* ‘Evora’, and *Festuca rubra trichophylla* ‘Smyrna’) exhibited a progressive decrease in total chlorophyll content as salinity conditions became more intense [24,32]. In addition to decreased chlorophyll content, salt stress has been reported to cause chloroplast structural damage. As documented in tall fescue and many other higher plants subjected to salinity stress, the arrangement of chloroplasts in mesophyll cells may have become disordered, the connection between grana loosened, and the lipid-bilayer of chloroplasts were damaged [33,34].

Cellular membrane structural damage and lipid peroxidation are common responses to oxidative stress. Many, if not all, edaphic stresses on turfgrasses cause increased production and accumulation of reactive oxygen species. The presence of a higher concentration of reactive oxygen species can cause further plant damage and even death by inactivating certain proteins and enzymes and destroying cell membrane structure and permeability via lipid peroxidation. Plants defend against reactive oxygen species through antioxidant defense systems that respond to an environmental stress. Stimulated activity of certain antioxidant enzymes, such as superoxide dismutase, catalase, hydrogen peroxidase, and ascorbate peroxidase, is used to scavenge reactive oxygen species to reduce oxidative damage. For example, superoxide dismutase converts oxygen radicals into hydrogen peroxide, which is then scavenged into molecular oxygen and water [35,36]. Antioxidant enzyme activity under saline conditions has been described for some turfgrass species. Kurup et al. [36] measured leaf firing percentage (visual turf quality), and activity of superoxide dismutase, catalase, hydrogen peroxidase, and ascorbate peroxidase of three seashore paspalum genotypes, four bermudagrass cultivars, and tall fescue at five different salinity levels (0, 5000, 10,000, 15,000, and 20,000 mg L^−1^ NaCl). Results indicate that increased antioxidant enzyme activity in a genotype correlated to a higher salinity tolerance through a lower leaf firing %. Bermudagrass cvs. ‘Princess 77′ and ‘Tifgreen’ as well as tall fescue were shown to be the least salinity tolerant throughout all parameters. These cultivars exhibited the highest leaf firing at all salinity levels, and bermudagrass cvs. exhibited amongst the lowest activity for all enzymes at the highest salinity level. All seashore paspalum cultivars (‘Salam’, ‘Seadwarf’, and ‘Seaisle 2000′) exhibited minimal leaf firing at all salinity levels when compared to bermudagrass cvs. ‘Salam’ showed the lowest leaf firing % of paspalum genotypes and the highest hydrogen peroxide activity [36]. These results suggest that any one of the tested paspalum genotypes is viable for high salinity conditions.

Due to enhanced nutrient competition as a result of excessive NaCl uptake in saline conditions, ion imbalances and toxicities are common effects of salt stress on plants. Sodium (Na^+^), chloride (Cl^−^), and boron (B^3+^) toxicities pose the greatest threat to plant growth, whereas deficiencies of calcium (Ca^2+^), potassium (K^+^), and iron (Fe^2+^) also create adverse effects on plant growth while under salt stress.

### 3.1. Sodium (Na^+^) Toxicity 

In saline soils, sodium is the most common toxic element due to the large amounts of soluble salts in the soil solution. High concentrations of sodium in plant tissue negatively impact cell division and enzyme activity through a variety of mechanisms such as disturbance in osmoregulation; uptake of other ions such as K^+^, Ca^2+^, and NO_3_^−^; and increased production of reactive oxygen species [37]. Such mechanisms lead to a decrease in chlorophyll content and destruction of the cellular membrane and organelle structure, thus impeding plant growth and development at all stages [38]. Guo et al. [39] found that no correlation was found between leaf firing % and Na^+^ shoot content while under salt stress. However, a significant negative correlation existed between the Na^+^ shoot content and the relative shoot weight and relative root weight while under salt stress [39]. The results of this study indicate that sodium accumulation in leaf tissue does not have a direct impact on leaf firing/scorch, which is a common symptom associated with salt stress. As a result of increasing NaCl content in the plant and the soil, K^+^ and Ca^2+^ often become deficient and may restrict plant growth. A large external influx of Na^+^ inhibits root uptake and decreases the relative binding affinity of K^+^ and Ca^2+^ for binding sites on enzymes due to their similar hydrated radii and hydration energy to Na^+^ [38].

Salts used in road de-icing during the winter season inhibit the growth and development of utility turfgrass species. Germination capacity was evaluated at five salt solutions of NaCl (0, 50, 100, 150 and 200 mM), and physiological parameters were measured during the tillering phase at salinity levels of 0, 150, and 300 mM of NaCl [40]. Seeds of Kentucky bluegrass were found to have no germination under any level of salinity. During the tillering phase, salinity negatively affected the length, area, and dry mass of roots as well as the relative water content of plants. The maximum chlorophyll fluorescence yield, quantum yield of photosystem II, and electron transport rate were negatively impacted at the early period of stress [40].

### 3.2. Chloride (Cl^−^) Toxicity

Both boron and chlorine are considered essential plant micronutrients for plant growth and metabolism; however, in some instances they can become toxic to plants either naturally or anthropogenically. Chlorine toxicity, as well as most micronutrients such as boron and zinc, most commonly occur when using recycled wastewater on soils with low percolation rates in arid and low-rainfall areas due to the ions’ high mobility in soil [41]. As an anion, chloride plays essential roles in stomatal regulation, photosynthetic activity, and osmoregulation. Stomatal regulation is mediated by the fluxes of potassium and accompanying anions such as chloride to act as a stabilizer. Chloride (Cl^−^) and manganese (Mn^2+^) work harmoniously in the water splitting reaction, or Hill reaction, in photosystem II. Excessive accumulation of chloride in the chloroplasts can cause chlorophyll degradation through a chloride-induced disturbance of photosystem II’s structural organization because of the highly permeable chloroplast envelope [42,43]. Perhaps most importantly, chloride helps regulate osmotic pressure in vacuoles at higher concentrations and more specialized tissues, such as apical meristems and root tips, when at low concentrations [44]. However, too much Cl^−^ can accumulate in leaf tissue, resulting in a bleached or burned appearance of mature leaf tips and margins, thereby negatively impacting photosynthetic efficiency and potentially decreasing a species’ salt tolerance by disrupting regular osmoregulatory practices [41].

When two popular C_3_ cool-season turfgrass species of Kentucky bluegrass (*Poa pratensis* L.) and tall fescue (*Festuca arundinacea* Schreb.) were compared, Kentucky bluegrass accumulated more Na^+^ and Cl^−^ in its shoots, although Na^+^ and Cl^−^ concentrations in shoots and roots increased with increasing salinity in both turfgrasses treated with 0, 50, 100, 150, and 200 mM NaCl growing for 40 days [45]. The total soluble sugar (TSS) concentration of the tall fescue was significantly higher than that of the Kentucky bluegrass under elevated NaCl concentrations, with significantly lower nitrate (NO_3_^−^) concentrations in the shoots of the Kentucky bluegrass, which indicate either poor nitrate transportation from roots to shoots or a stronger negative competition between the two anion concentrations of nitrate and chloride in Kentucky bluegrass [45].

### 3.3. Potassium (K^+^) Deficiency 

Potassium plays important roles in stomatal regulation, CO_2_ fixation, and the detoxification of reactive oxygen species [46]. With adequate levels of K, CO_2_ fixation is increased, and translocation of photosynthates from source to sink organs is also increased to inhibit the photosynthetic electron transport to O_2_ to form excess reactive oxygen species [47,48]. Potassium is also shown to downregulate the activity of NADPH oxidase that converts NADPH to a superoxide radical and NADP^+^ [49]. Therefore, potassium-deficient plants will show reduced rates of photosynthesis followed by damaged membranes and chlorophyll structures due to the overproduction of ROS [46,50]. *Puccinellia tenuiflora* is a saline–alkali-tolerant plant in the Songnen Plain, one of the three largest soda saline–alkali lands worldwide. The soils from the Songnen Plain were reasonably rich in salts and alkali, and the soils were severely deficient in nitrogen, phosphorus, and potassium. When investigated for its ability to tolerate these adverse soil conditions, unigenes involved in the uptake of N, P, K, and micronutrients were found to be significantly upregulated in the soil, which indicated the existence of an efficient nutrition-uptake system in *P. tenuiflora.* Compared with *P. tenuiflora*, rice (*Oryza sativa*) was hypersensitive to saline–alkali stress [51].

In a 2021 study, perennial ryegrass (*Lolium perenne* L.) seeds were cultured in soil mixtures amended with clinoptilolite zeolite (OZ) and potassium-enriched clinoptilolite zeolite (K-EZ), then exposed to three salinity levels (0, 50, or 100 mM NaCl) for three months. The results show that the application of both types of zeolite significantly decreased Na content by 44.36% and 21.31%, but increased K content by 272.34% and 81.59%, as well as the K/Na ratio by 590.47% and 129.43%, in shoots and roots, respectively. Si content in shoots was increased by 28.33%. The soil mixtures with zeolite, especially K-EZ, enhanced relative water content, membrane stability index, total chlorophyll content, total soluble proteins, peroxidase, and superoxide dismutase activities, but reduced the contents of total soluble carbohydrates, hydrogen peroxide, and malondialdehyde in saline conditions. Shoot and root dry weight, root volume, and root/shoot ratio were also found to be improved by the application of OZ and K-EZ [52].

### 3.4. Calcium (Ca^2+^) Deficiency 

Calcium is an important factor for cell wall and membrane stability, as well as serving as a secondary messenger to both biotic and abiotic stresses [53]. Ca^2+^ signals help coordinate plant responses and defense mechanisms that are induced by many edaphic stresses including salt stress [54]. This would suggest that deficient levels of Ca decrease the plants’ ability to sense and tolerate salt stress, leading to more significant plant damage. Since calcium-deficient plants lack the ability to inhibit salt translocation to the shoots, shoot tip injury, loss of apical dominance, and the development of axillary shoots with necrotic young leaves are common Ca-deficiency symptoms [50].

Tall fescue, after a three-month establishment of canopy and roots, was treated with a combination of 300 mM NaCl and calcium nitrate for five days. The nitrogen content in shoots was highly correlated with Ca^2+^ and K^+^ contents in roots. Additionally, high levels of salt increased ATP6E and CAMK2 transcription levels in shoots at day one and day five treated with calcium nitrate. In roots, CAMK2 level was reduced by salinity at day five and exogenous calcium helped to recover it. These findings indicate exogenous calcium plays positive roles in tall fescue to improve salt tolerance [55].

### 3.5. Boron (B^3+^) Toxicity 

Boron’s primary function in plants is to help maintain cell wall structure and function. Borate (boric acid) forms complexes with sugars that stabilize the pectin network to regulate cell wall pore sizes. Boron also participates in metabolic processes that influence vegetative and reproductive growth, such as the metabolism of nucleic acids, phosphorus, nitrogenous compounds, and hormonal regulation [56]. Similar to chloride (Cl^−^), excessive accumulation of boron in leaf tissue causes leaf scorch appearing as necrotic and/or chlorotic tissue on the tips and margins of mature leaves [56]. The accumulation of high concentrations of boron can cause osmotic imbalances, therefore reducing the plants’ ability to tolerate oxidative damage leading to lipid peroxidation, increases in membrane permeability, and accumulation of proline [56,57]. These responses to boron toxicity can inhibit photosynthesis by damaging thylakoid formation and structure, thus reducing CO_2_ absorption and creating conditions that disrupt the transmission of photosynthetic electrons and molecular oxygen to act as an electron acceptor to overproduce reactive oxygen species [57]. Boron toxicities have also been shown to create physiological effects during plant growth, such as reducing root growth in rice and inhibiting seed germination in various plant species [58,59]. This growth and development inhibition is proposed to be caused by boron’s ability to bind to the ribose section of several key metabolites such as ATP, NADPH, and NADH [57].

Boron toxicity occurs in some salt-affected soils. This is often observed in arid or semi-arid soils and when the irrigation water is high in salt contents [60]. Buffalograss (*Buchloe dactyloides* (Nutt.) Engelm.) pots in a greenhouse were irrigated with solutions containing 0.5, 1, 2, 4, 6, 8, or 12 mM of boron (B), chlorine (Cl), copper (Cu), iron (Fe), manganese (Mn), molybdenum (Mo), and zinc (Zn) [61]. Boron and Mo induced visual toxicity symptoms more readily than other micronutrients. Boron toxicity of chlorosis was often accompanied by bleached leaf tips. Biomass yield was reduced when the nutrient solution contained 2 mM B, 6 mM Cu, or 2 mM Mo, although the elevated levels of other micronutrients of Cl, Fe, Mn, and Zn did not alter dry matter yield [61].

## 4. Turfgrass Salinity Resistance Mechanisms

Since salinity resistance of the *Poaceae* family encompasses a broad spectrum of salinity levels, there are a multitude of salinity resistance mechanisms that turfgrass species use (Figure 3). Plant salinity resistance is a combination of two processes: avoidance and tolerance. Avoidance mechanisms aim to decrease excessive translocation of salts to the more sensitive leaf tissues, while tolerance mechanisms aim to increase the plant’s ability to survive in the presence of accumulated salts in leaf tissues [62]. This is done through a variety of physiological adaptations such as stimulated root growth, osmotic adjustment and ion exclusion, ion regulation, ion sequestration, and ion excretion. All resistance mechanisms can occur at the same time within a species and are shared across most salinity-tolerant plant species [4].

### 4.1. Turfgrass Morphological and Anatomical Resistance Mechanisms

Most morphological plant responses to salinity stress occur in the form of leaf scorch and wilting; however, root growth can also be positively or negatively affected by saline conditions. Salt-tolerant turfgrasses often exhibit stimulated root growth under low to moderate salinity levels. Stimulated rooting has been observed in many warm-season moderately salt-tolerant and halophytic turfgrasses [21,63,64]. This mechanism increases the root absorptive area/shoot transpirational area ratio by both increasing root biomass and decreasing or inhibiting relative shoot growth. More root biomass helps counter external osmotic stress and overaccumulation of salts in leaf tissue by allocating more biomass that can safely accumulate salts with minimal adverse effects. This has been observed in *Cynodon dactylon* × C. *transvaalensis* cv. ‘Tifway’, *Cynodon dactylon* cv. ‘Celebration’, *Stenotaphrum secundatum* cv. ‘Raleigh’, *Paspalum vaginatum* cv. ‘SeaStar’, and *Zoysia japonica* cv. ‘Palisades’ increased their root biomass by 59%, 29%, 113%, 24%, and 27% compared to the control (2.5 dS m^−1^), respectively, when exposed to 15 dS m^−1^ salinity level for two years [63]. Most cultivars that exhibited stimulated root growth maintained or increased aboveground visual turf quality (TQ); however, *Cynodon dactylon* × *C. transvaalensis* cv. ‘Tifway’ had a significant decrease in TQ over the two-year period suggesting that ‘Tifway’ relies on stimulated root growth as its primary mechanism for salt tolerance rather than ion exclusion or regulation [63].

Salt-sensitive plants over-accumulate salts in the plant tissues beyond the amount needed for osmotic adjustment. This, in effect, raises the leaf osmolarity well beyond the root-media salinity, causing toxicity symptoms such as reduced growth and leaf scorch. Salt-tolerant plants maintain cell turgor pressure and normal physiological functions during salt stress by sufficiently increasing sap osmolarity in the xylem to compensate for external osmotic stress, a process known as osmotic adjustment [65]. However, loading of sodium ions into the xylem and vacuoles are active processes due to the membrane negative charges. Therefore, osmotic adjustment is a relatively inefficient avoidance mechanism in many non-halophytic species, because of higher leakage rates across the tonoplast, when compared to halophytic species [66]. Shoot saline ion exclusion paired with osmotic adjustment has been shown to be perhaps the most prevalent mechanism in salt avoidance in both C_3_ and C_4_ turfgrass species [66,67,68]. Ion exclusion refers to the exclusion of saline ions from the shoots to minimize toxic effects by inhibiting translocation of Na^+^ to the shoots [69]. Many studies have shown that shoot concentration of Na^+^ and Cl^−^ is negatively associated with relative salinity tolerance of both C_3_ and C_4_ turfgrass species [33,62,64]. This correlation has been successfully used to predict salinity tolerance of *Cynodon* spp. and *Zoysia* spp. [68,70]. Multiple mechanisms of sodium exclusion have been studied, including Na^+^ efflux from roots, Na^+^ partitioning in vacuoles of root cells or mesophyll cells, and control of Na^+^ loading and unloading by xylem parenchyma cells [69]. Salt-tolerant plant species have been reported to have higher vacuolar and plasma membrane H-ATPase enzyme activity, which is highly correlated with the rate of root Na^+^ efflux, when compared to salt sensitive cultivars [71,72]. Higher H-ATPase enzyme activity was shown to generate an electrochemical gradient as the driving force for Na^+^ exclusion [73]. The ability of a plant species to control xylem ion loading is one of the most important mechanisms for reducing Na^+^ transport to shoots [69]. Multiple passive and active mechanisms are proposed to be responsible for Na^+^ loading at the xylem/parenchyma interface, including the salt over sensitive 1-encoded Na^+^/H^+^ antiporter pathway, the high affinity K^+^ transporter pathway, and the cation-chloride cotransporter [74,75].

### 4.2. Sodium and Potassium Uptake Mechanisms

More recently, salinity tolerance has been suggested to be associated more with the ion-specific component rather than with the osmotic component of stress [62]. Regulation or selectivity of ion uptake, such as Na^+^, K^+^, and Cl^−^, are crucial for the functionality of osmotic adjustment because of their commonality in saline solutions. Higher salt tolerance is positively correlated with a plant’s ability to maintain a high K^+^/Na^+^ ratio when under salt stress [62,64]. This occurs by reducing the transport of Na^+^ from the roots to the shoots, increasing the absorption of K^+^ in the roots and reducing the leakage of K^+^ from cells [64]. Ion regulation primarily occurs at the casparian strip of the root endodermis and low-affinity K^+^ channels in plasma membrane of root cells [76]. Maintenance of a high K^+^/Na^+^ ratio in the cytoplasm is mediated by Na^+^/K^+^ antiport activity, which further aids in proper enzyme function, cell metabolism, and photosynthetic pathways [69]. In salt-tressed seashore paspalum, the K^+^ concentration decreased by 26.20% in the shoots and 69.68% in the roots; however, the Na^+^ concentration increased 15-fold in the shoots and 25-fold in the roots. Possibly, the regulation mechanisms of the K^+^ and Na^+^ for seashore paspalum allow the maintenance of high K^+^ concentrations in the shoots to decrease Na^+^ translocation from the roots to the shoots under high salinity stress [33]. A significant negative correlation was found between the K^+^ concentration in the roots and the Na^+^ concentration in the shoots under salinity stress. This means that more K^+^ were taken up by roots, there was higher K^+^/Na+ in the roots, and that more K^+^ were transferred from roots to shoots, so the transfer of Na^+^ from roots to shoots was inhibited [33]. An overexpression of MicroRNA393 in transgenic creeping bentgrass (*Agrostis stolonifera*) led to significantly higher accumulation of Na in the shoots and K in both the roots and shoots while exposed to 200 mM of NaCl. Despite the high accumulation of Na in the shoots of the transgenic lines, the increased accumulation of K in the shoots helped mitigate further oxidative damage and improve membrane stability in the salt-stressed transgenic lines. This was reflected in an increased relative water content and decreased electrolyte leakage when compared to the wild-type control lines. That said, upon further research, —the authors found that an overexpression of MicroRNA393 aids in maintaining membrane stability and chlorophyll content while under salt stress, thus resulting in less leaf damage and a higher salinity tolerance [77]. It is also proposed that calcium ions play an important role in signaling ion regulation during salt stress. In response to NaCl exposure at the roots, Ca reporter proteins trigger systemic, wave-like fluctuations of cytosolic Ca^2+^ content, which is decoded into downstream responses. This is described as the salt overly sensitive1 signaling pathway. Salt overly sensitive3 is a calcium binding protein that interacts with and activates the CBL-interacting protein kinase. The resulting Ca^2+^ sensor-kinase complex then phosphorylates and activates the Na^+^/H^+^ antiporter salt overly sensitive1, which functions in Na^+^ extrusion and long-distance Na^+^ transport in plants. In addition, the Ca^2+^ sensor-kinase complex also regulates ion homeostasis and nutrient uptake by activating specific ion transporters and channels [78].

Ion sequestration or compartmentation is a tolerance mechanism that restricts the number of ions in the cytoplasm by sequestering saline ions in the vacuole. When under high saline conditions, specific types of organic solutes called ’compatible solutes’ are accumulated in the cytoplasm to promote saline ion translocation across the tonoplast and into the vacuole. The accumulation of saline ions is metabolically expensive and can inhibit enzyme activity, which would decrease ion movement across the tonoplast. ’Compatible solutes’, such as proline and glycine betaine, are accumulated in the cytoplasm under saline conditions to sufficiently maintain cellular structure and osmotic balance. Both proline and glycine betaine have been shown to be positively correlated with salinity tolerance by regulating ion homeostasis, enhancing osmotic adjustment, and scavenging reactive oxygen species [62,79]. Proline’s primary function is to increase antioxidant activity to mitigate lipid and protein peroxidation. Proline also improves membrane stability and contributes to intracellular osmotic adjustment to promote ion sequestration and maintain photosynthetic activity while under salt stress [62,80]. In rice, the overexpression of transcription factor OsMADS25 resulted in a greater accumulation of proline in the roots causing an increased carbohydrate content and higher salinity tolerance when compared to the wild type [80]. Similar plant responses to glycine betaine have been reported to improve salt tolerance of certain plant species. In wheat, through the introduction of betaine aldehyde dehydrogenase, transgenic lines overexpressing glycine betaine may elicit enhanced salt tolerance. Glycine betaine was shown to have a protective effect on the components and function of photosystem II (PSII) within the thylakoid membrane, which was reflected in a higher photosynthetic rate when compared to the wild types [79]. Exogenous glycine betaine enhanced the salt tolerance in perennial ryegrass (*Lolium perenne*) through elevated antioxidant enzyme activity that decreased cell membranes and improved ion balances by maintaining a higher K/Na shoot ratio while under salt stress [81].

### 4.3. Ionic Excretions

Ion excretion is a salinity tolerance mechanism in many higher plant species that promotes active excretion of excess saline ions from the shoots via external storage structures on the leaf epidermis. Bicellular leaf epidermal salt glands are responsible for ion excretion in turfgrass species in the *Chloridoideae* subfamily including *Bouteloua* spp., *Buchloe dactyloides*, *Cynodon* spp., *Distichlis spicata* ssp. stricta, *Sporobolus virginicus*, *Zoysia matrella*, and *Z. japonica* [5]. Salt glands have been found on both abaxial and adaxial leaf surfaces and are arranged longitudinally in parallel rows adjacent to rows of stomata [82,83]. A cutinized basal cell on the leaf epidermis and a cap cell comprises the salt glands. Basal cell appearance varies amongst species, from being fully embedded to fully extruded from the leaf epidermis [5,83]. Multiple structural modifications of the leaf epidermis have been proposed to be responsible for salt partitioning and excretion. A series of parallel invaginated channels going in the direction of ion flow on the plasma membrane (infoldings of the plasmalemma) of the basal cell are suggested to partition and guide saline ions into the salt glands on the leaf epidermis [83]. In addition, the localization of ATPase enzyme activity on the salt gland basal cells suggests that there is also active ion loading occurring at the salt glands to promote excretion [84]. Studies showed that ‘Celebration’ bermudagrass (*Cynodon dactylon*) and ‘DALZ1313′ zoysiagrass (*Zoysia matrella* × *Z. japonica*) each developed salt glands on the adaxial leaf surface at the 30 dS m^−1^ salinity level, while DALZ1313 also produced salt glands under control conditions (2.5 dS m^−1^). In addition, salt crystals were observed on leaf surfaces, indicating that ion excretion was present in both species. Salt gland density increased with increasing salinity (from 2.5 to 30 dS m^−1^) for DALZ1313 in response to increased salinity, suggesting that increased salt gland density could be one of several responses contributing to enhanced salinity tolerance [5]. However, other studies have indicated *Zoysia matrella* showing no difference in salt gland density and size with increasing salinity levels [85]. This suggests that responses of salt gland density are variable amongst the cultivar and species level. Several environmental factors are suggested to affect the rate of salt secretion in all plant species, but these are not fully understood as results often conflict with each other between species [83]. This would suggest that there is a great number of differences across species in regulation of salt secretion. Many studies indicate that salt excretion is influenced by solar radiation and the rate of salt translocation in plants, suggesting that salt excretion rates maximize during the day [86]. It is proposed that saline ion concentrations in soil significantly affect excretion rates. Sugiura determined that the salt glands of *Zoysia matrella* demonstrated that Na^+^ excretion increased with increasing salt treatment concentrations [85]. Due to the relation of osmolarity and soil moisture, Sugiura’s finding suggests that for some species, salt gland excretion rate may be more dependent on soil moisture status. Furthermore, this causes excretion rates to be variable throughout the day and amongst species, depending on their respective photosynthetic pathway. Other mechanisms of ion excretion within the *Poaceae* family have been studied such as bladder-like structures exclusive to *Paspalum vaginatum*, the most tolerant warm-season perennial turfgrass species. Salt bladders or papillae are unicellular structures while salt glands are bicellular, otherwise the two structures are oriented and function similarly [87]. Seashore paspalum exhibited bladder-like structures under both control and 30 dS m^−1^ salinity, which developed along the vascular bundles on the adaxial leaf surface [5]. Another study indicated dense ridges of papillae on the adaxial leaf surface of seashore paspalum [87]. Differences were observed between tested cultivars. Cultivars ‘HI10′ and ‘509018-3′ were the more salt-tolerant cultivars, exhibiting larger papillae and a higher concentration of Na^+^ in the papillae than in the underlying leaf tissue, resulting in a higher K concentration in the leaf tissue when compared to other cultivars [87].

## 5. Management Practices to Improve Turfgrass Salinity Resistance

Various management and cultural practices may be valuable efforts in enhancing turfgrass salinity resistance without using a potable water source. Cultural practices include selecting a more salt-tolerant turfgrass species or cultivar and increasing the mowing height, which can enhance turfgrass salt tolerance. Management practices such as applications of some soil amendments, silicon, and plant growth regulators have also shown positive effects on saline soil recovery and plant salt tolerance.

### 5.1. Turfgrass Species and Cultivar Salinity Tolerance Variations

Most turfgrass species fall within the range of slightly tolerant to tolerant categories of relative salinity tolerances. With each species, multiple genetic lines, varieties, and cultivars exist and provide a greater opportunity to select a proper turfgrass for a particular site (Table 2). Proper species and cultivar selection for use in saline soil or when using saline irrigation water is of utmost importance when considering maintenance costs and turf quality. Turfgrasses have great species and varietal differences when it comes to salinity resistance. The majority of cool-season turfgrass species and cultivars are defined as glycophytes, which cannot grow in high salt levels. However, in lower saline conditions, glycophytes can be considered sensitive or moderately sensitive to salt stress by triggering a variety of resistance mechanisms [88]. Some plant and turfgrass species are halophytic in nature, as discussed earlier, meaning that they can withstand higher soil salinity levels up to and beyond that of seawater without experiencing severe reductions in growth or metabolic function [28]. Some examples of halophytic turfgrass species include alkaligrasses (*Puccinellia* spp.), inland saltgrass (*Distichlis spicata* sp. stricta), and seashore paspalum (*Paspalum vaginatum*). Although halophyte and glycophyte classifications provide a general level of salt resistance to certain species, the degree of salinity impact on turfgrasses varies among species and the level of genetic tolerance at the cultivar level [19]. As with any stress tolerance, it is difficult to determine the absolute salinity resistance of a genotype and compare relative resistance between genotypes because of many other factors.

### 5.2. Environmental Factors

Environmental factors such as temperature and light greatly influence plant salinity resistance. Hot and dry climates are often associated with a greater degree of salinity stress because of the increased evapotranspirational demand that favors salt uptake when compared to cool and humid climates [89]. Thus, warm-season C_4_ turfgrass species are oftentimes more capable to adapt and survive salinity stress when compared to cool-season C_3_ species because of their natural habitats. Warm-season turfgrass species include *Cynodon dactylon* × C. *transvaalensis* cv. ‘Tifway’, *Cynodon dactylon* cv. ‘Celebration’, and *Paspalum vaginatum* cvs. ‘Sea Isle 1’ and ‘SeaStar’ maintained the highest normalized difference vegetation index, visual turf quality, lowest growth reduction, and lowest percentage of leaf firing under elevated salinity when compared to other species tested. Increased shoot growth and turf quality were noted in ‘Celebration’ bermudagrass and both seashore paspalum varieties at the 15 dS m^−1^ treatment in relation to the control treatment (2.5 dS m^−1^). However, most zoysiagrass (‘Palisades’ and ‘Zeon’) and St. Augustinegrass (‘Raleigh’, ‘Floratam’, and ‘Palmetto’) varieties exhibited decreased growth and turf quality in response to any magnitude of elevated salinity in relation to the control treatment. Both bermudagrass and seashore paspalum cultivars recovered back to an acceptable turf quality at the 15 dS m^−1^ treatment, while no cultivars tested in the study recovered to an acceptable level at salinity levels 30 dS m^−1^ and 45 dS m^−1^ [63,90]. A recent study evaluating the salinity tolerance of eight total varieties in four different C_3_ turfgrass species, indicated that *Lolium perenne* cv. ‘Ringles’, *Festuca rubra* L. ssp. *trichophylla* cv. ‘Abercharm’, *Poa pratensis* cv. ‘Prafin’, *F. arundinacea* cvs. ‘Fesnova’, and ‘Golden Gate’ demonstrated salt tolerance when treated with up to 200 mM of NaCl. These varieties exhibited a higher germination percentage, plant growth, photosynthetic efficiency, and K^+^/Na^+^ ratio under salt-stressed conditions when compared to the other varieties tested [26]. Significant reductions in turfgrass quality and development associated with gradually increasing salinity levels (from 0.54 mM to 200 mM NaCl) were noted for *Poa pratensis* cv. ‘Evora’, while *Festuca rubra commutata* cv. ‘Casanova’ was only moderately affected by salinity. *Lolium perenne* cv. ‘Belida’ and *Festuca rubra trichophylla* cv. ‘Smyrna’ had the least-affected clipping yield and total root and shoot dry weight values while increasing salinity and are thus considered highly tolerant to salinity stress [32]. When salinity tolerance of two cultivars of Kentucky bluegrass (*Poa pratensis*), ‘Limousine’ and ‘Kenblue’, were evaluated, ‘Limousine’ exhibited a 25% shoot growth reduction at 4.7 dS m^−1^ EC, while ‘Kenblue’ exhibited a 25% shoot growth reduction occurring at 3.2 dS m^−1^. ‘Limousine’ also maintained a higher K^+^/Na^+^ ratio and relative water content under moderate salinity (8.2 dS m^−1^), suggesting that *Poa pratensis* cv. ‘Limousine’ is more salt-resistant than Poa pratensis cv. Kenblue [91].

### 5.3. Practical Management Approaches

The most common remedy that crop producers and turfgrass managers use to improve soil structure and ameliorate soil salinity is the application of gypsum (CaSO_4_2H_2_O). Gypsum is a cheap and relatively easy to use soil amendment that has the potential to reduce soil pH, remove excess sodium ions from the soil’s cation exchange sites, and provide calcium to salt-stressed plants. The sulfur component in gypsum provides the ability to form strong acids in the soil, such as sulfuric acid (H_2_SO_4_), to lower the soil pH. The soil pH of a sandy loam soil when 4 tons ha^−1^ of gypsum was applied experienced a slight decrease over a two-year period [92]. In the same study, the effect that gypsum applications had on saline soil reclamation was greater when paired with a leaching factor. Electrical conductivity, sodium absorption ration, and exchangeable sodium percentage values demonstrated a decreasing trend as application rates of gypsum increased with a leaching factor. The highest crop yield of *Allium cepa* cv. ‘Bombe red’ was found in the treatment with the highest gypsum rate (4 tons ha^−1^) plus a leaching factor. Use of only a leaching factor provided non-significant results in saline soil reclamation when compared to treatments including both a leaching factor and gypsum [92]. The calcium in gypsum provides a stronger competition for Na^+^ on the soil cation exchange capacity sites, causing the Na^+^ to be replaced by the Ca^2+^ and the insoluble sodium sulfate (Na_2_SO_4_) to be leached out. This helps prevent soil deflocculation and unstable soil structures, which are common issues in saline soils [93]. Applications of gypsum on a mixture of sand-based soil media increased the soil moisture content and calcium content and decreased the sodium adsorption ratio of all soil media mixtures when compared to gypsum-free treatments. However, gypsum had no significant relationship with soil pH and clipping yield of Kentucky bluegrass (*Poa pratensis*) on any soil media [94]. Gypsum treatment had a significant effect on the salinity tolerance of ‘Zeon’ Zoysiagrass, ‘Platinum’ Seashore paspalum, and ‘TifEagle’ bermudagrass after 2 weeks of treatment. Electrolyte leakage of salt-stressed plants decreased with the application of gypsum in all tested turfgrass species [95]. Exogenous calcium supplementation in rice was found to reduce cadmium-induced oxidative stress through a reduction in hydrogen peroxide content, and an increased activity of antioxidant enzyme activity superoxide dismutase, catalase, and glutathione S-transferase when compared to cadmium treatments with no exogenous calcium applications. Furthermore, increased carotenoid content and visual rice quality was associated with calcium-treated rice at all levels of cadmium stress including the control [96]. An increase in calcium concentration in rhizosphere was also noted in Challa’s [92] study with applications of gypsum and leaching factors on onion crops, thereby reducing the Na uptake and increasing K^+^ uptake, which in turn caused an increased K^+^/Na^+^ ratio in roots while the turfgrass species were exposed to saline conditions [95].

Recent research has indicated that the use of bioorganic soil amendments has had positive effects on plant salinity tolerance and enhanced physical, chemical, and biological properties of saline soils. Applications of humic acid (0.84 and 2.54 L ha^−1^) on two golf course fairways, one in Colorado and the other in North Dakota, improved visual turf quality, organic matter content, and microbial biomass in both inherently saline soils and soils irrigated with recycled wastewater. However, humic acid did not significantly reduce soil pH or electrical conductivity over the two-year period but increased other aspects of the soil to yield a better-quality fairway [97]. The use of organic amendments on saline soils also improved many physical properties including porosity, hydraulic conductivity, and soil structure stability, which are all negatively affected by soil salinity [93,97]. Soil microbes are extensively associated with nutrient availability through mineralization and immobilization, thus potentially improving plant tolerance and soil fertility under saline conditions [93]. Plant-growth-promoting rhizobacteria have various direct and indirect mechanisms to mitigate plant growth and quality reduction in responses to salt stress [98,99,100]. A salt-sensitive rice cultivar (*Oryza sativa* cv. ‘GJ17′) that was inoculated with the plant-growth-promoting rhizobacterium species *Pseudomonas pseudoalcaligenes* and *Bacillus pumilus* (independently and separately) experienced a reduction in the harmful effects of salinity stress. Treatments including plant-growth-promoting rhizobacteria resulted in increased plant biomass amongst all treatments, and the treatment with both plant-growth-promoting rhizobacterium species exhibited a synergistic effect to create the greatest biomass. Inoculation of ‘GJ17′ with plant-growth-promoting rhizobacteria reduced the toxicity of reactive oxygen species on plant cells, as reflected through a decrease in lipid peroxidation and caspase-like protease activity. In effect, this enhanced membrane stability, decreased programmed cell death, and increased cell viability of the plants when under salt stress [99].

Salt stress originates from saline solutions remaining in the root zones for a sufficient period for plants to accumulate salts to a toxic level. Therefore, sufficient leaching and adequate drainage are critical when using reclaimed water as an irrigation source [65,101]. Highly maintained turfgrass surfaces such as golf course greens, tees, and fairways are highly prone to salinity stress due to the low mowing height. Since salinity stress largely affects and accumulates salts within leaf tissues, turfgrasses with lower shoot biomasses can exhibit more severe salinity stress symptoms when compared to taller cut landscapes [102,103,104]. As the mowing heights increased from 15 mm to 45 mm of three bermudagrass cvs. (‘Tifway’, ‘Tifgreen’, and ‘Tifdwarf’), higher turf quality was noted in the higher-cut treatments (45 mm) when exposed to a 16 dS/m salinity level across all cultivars. In addition to higher turf quality with increased mowing heights, higher photosynthetic rates and total nonstructural carbohydrate contents were also responsible for increased salt resistance of bermudagrass under salinity stress [103]. Lower mowing heights of ‘L-93′ creeping bentgrass (*Agrostis stolonifera*) putting greens were reported to severely decrease in salinity resistance as determined by visual turf quality. When mowed at 6.4, 12.7, and 25.4 mm the turfgrass reached an unacceptable quality at soil EC 4.1, 12.5, and 13.9 dS/m, respectively [102]. The previous studies suggest that a higher shoot biomass increases turfgrass salinity tolerance by providing more plant tissue to sequester or excrete saline ions and produce more photosynthates.

Silicon has become a renowned beneficial macronutrient that can help mitigate a wide variety of biotic and abiotic stresses [105]. Silicon has been reported to regulate carbohydrate metabolism to improve plant yield and photosystem II photochemical efficiency under salinity stress [106,107]. Chlorophyll content was significantly increased in both salt sensitive species of turfgrass, *Lolium perenne* and *Festuca arundinacea* when silicon was applied while under salt stress, while more salt resistant *Cynodon dactylon* only exhibited a slight increase in chlorophyll content [108]. This would suggest that salt sensitive plant species may benefit from exogenous silicon applications greater than salt resistant species. The mechanisms in which silicon increases photosynthesis and plant growth under abiotic stresses is relatively unknown. However, silicon’s effects on ion uptake and oxidative stress have been studied intensively. In the same study, silicon treatments reduced both shoot and root Na^+^ concentrations in *Cynodon dactylon, Lolium perenne*, and *Festuca arundinacea*, while no significant increases in root or shoot potassium content were noted across all species [108] indicating that silicon may influence plant ion uptake to improve salinity resistance. Application of silicon overall reduced root and shoot concentration of Na^+^ in wheat. The silicon reduced root absorption and translocation to the shoots of Na^+^ resulting in a decrease in Na^+^/K^+^ ratio to mitigate salinity stress [109]. The same mechanism of silicon induced salt stress mitigation was found in maize with applications of 1 mM of Si, in the form of Na_2_SiO_3_. The applied Si alleviated salt stress through depreciations in Na^+^/K^+^ ratio, Na^+^ ion uptake at the surface of maize roots, and translocation in plant tissues with reduced Na^+^ accumulation in leaf tissues [110]. The mechanism in which the silicon reduced Na^+^ uptake and transport within the plant is suggested to be a silicon induced stimulation of H^+^-ATPase enzymes on the root plasma membranes [111]. H^+^-ATPase H^+^-pyrophosphatase enzymes play a crucial role in creating a potential gradient to power the tonoplastic Na^+^/H^+^ antiport that transfers Na^+^ from the cytoplasm to the vacuole [112]. Exogenous silicon applications have enhanced antioxidant enzyme activities and decreased oxidative stress in many plant species [113]. In wheat, applications of silicon at 5 mM and 30 mM increased antioxidant enzyme activity of both superoxide dismutase and catalase. Resulting in more efficient reactive oxygen species scavenging and reduced oxidative stress while exposed to 7.6 dS m^−1^ salinity level in the field [106]. Three turfgrass species (*Cynodon dactylon*, *Lolium perenne*, and *Festuca arundinacea*) demonstrated a similar effect on reducing oxidative stress through maintaining a higher relative water content and mitigating electrolyte leakage [108].

**Table 3 plants-12-00925-t003:** Some trials on mitigation of salinity, heat, and drought stresses by exogenous applications of plant metabolites among turfgrass species.

Exogenous Applications	Turfgrasses Tested	The Rates Used	Salinity Stress Level Range Used	Findings and References
24-epibrassinolide (EBL)-a plant hormone	Perennial ryegrass Tall fescue	0.15 mg L^−1^ EBL in 100 mL of the solution per pot (18 cm diameter and 20 cm depth)	0–6 dS m^−1^	The EBL application under salt stress alleviated loss in clipping yield by 35% and 12%; and reduced leaf firing, through weeks 2–6 post-application, by75–40% and 50–20% for perennial ryegrass and tall fescue, respectively [114].
Chitosan (CTS)-a natural polysaccharide	Creeping bentgrass	0.1, 0.2, 0.5, 1, and 2 g L^−1^	0–200 mM NaCl	The application of CTS increased antioxidant enzyme activities, thereby reducing oxidative damage to roots and leaves [115].
Glycinebetaine (GB)-an osmoprotectant	Creeping bentgrass Kentucky bluegrass Perennial ryegrass Tall fescue	50, 100, 150, or 200 mM solution of GB of seed priming	0.1 and 14.6 dS m^−1^	Seeds primed with GB showed a higher germination rate (11.0% to 13.9% increase) and seedling growth (19.3% to 20.7% increase) in mannitol or NaCl solution than in distilled water [116].
Melatonin (ME)-a natural hormone	Creeping bentgrass	0 or 20 μM ME	Drought stress	ME-alleviation of drought-induced leaf senescence in creeping bentgrass was associated with the down-regulation of chlorophyll catabolism and the synergistically interaction with CK-synthesis gene and signaling pathways [117].
Proline-a amino acid	Creeping bentgrass	Proline (10 mM)	Heat stress	Proline-enhanced heat tolerance of creeping bentgrass [118].
Spermidine (Spd)-a polyamine compound	Zoysiagrass (*Zoysia japonica* Steud) cultivars, ‘Z081′ and ‘Z057′	Spd 0, 0.15, 0.30, 0.45, 0.60 mM	0–200 mM NaCl	H_2_O_2_ and malondialdehyde (MDA) levels significantly decreased in both cultivars treated with Spd [119].
γ-Aminobutyric acid (GABA)-a neurotransmitter, a chemical messenger in human brain	Creeping bentgrass cv. Penncross	0.5 mM GABA	0–250 mM NaCl	GABA application is an efficient approach to enhance salt tolerance of creeping bentgrass during a prolonged period of salt stress and also provides valuable information to better understand key candidate genes and regulatory pathways of GABA-induced salt tolerance in plants [120].

### 5.4. Exogenous Applications of Plant Metabolites to Turfgrass Species

For many years, several groups of plant growth regulators have been used for improving turf quality and enhancing environmental stress tolerances. Propiconozole, a triazole fungicide that now can be used as a plant growth regulator, has shown to ameliorate salt stress in many plant species including annual vinca (*Catharanthus roseus*), perennial ryegrass (*Lolium perenne*), and Kentucky bluegrass (*Poa pratensis* cv. ‘Plush’) [121,122,123]. Annual vinca (*Catharanthus roseus*) exhibited higher root biomass and antioxidant enzyme activity under a salinity and propiconozole treatment (80 mM NaCl and 20 mg propiconozole) when compared to control (80 mM NaCl). Although the propiconozole caused reduced stem length and leaf surface area, applications of plant growth regulators on annual vinca enhanced salinity tolerance [121]. In addition to propiconozole, a fortified seaweed extract of seaweed (*Ascophyllum nodosum*), humic acid, thiamin, and L-ascorbic acid were shown to enhance salinity tolerance of perennial ryegrass. In Yan’s [123] dissertation, applications of both propiconozole and fortified seaweed extract had positive effects on leaf water potential, total lipid concentration, and nutrient content in perennial ryegrass while under salt stress. Propiconozole provided the greatest leaf water content and total lipid concentration, and fortified seaweed extract provided the greatest decrease in Na^+^ and Cl^−^ leaf tissue concentrations while under salt stress [123]. These results suggest that both propiconozole and fortified seaweed extract can be used to enhance salinity tolerance in turfgrass species through reductions in osmotic pressure and lipid peroxidation. A similar enhancement of salinity tolerance by foliar seaweed extract applications was noted on seashore paspalum (*Paspalum vaginatum* cv. ‘Salam’). While under saline conditions, seashore paspalum treated with seaweed extractant exhibited higher turf quality, K and Ca leaf content, photochemical efficiency, and antioxidant enzyme activity when compared to plants not treated with seaweed extractant [124]. In both studies, applications of fortified seaweed extract and propiconozole provided greater benefits only when the plant was salt-stressed [123,124]. Another plant growth regulator researched for turfgrass salinity tolerance is aminoethoxyvinylglycine, which is an ethylene synthesis inhibitor commonly used on fruit to delay ripening [125,126]. Drake’s [125] dissertation studied the effects of many plant growth regulator groups, including aminoethoxyvinylglycine, on creeping bentgrass’ (*Agrostis stolonifera*) salt tolerance. Applications of aminoethoxyvinylglycine at 10 µM exhibited the highest fresh root weight, tiller count, and total chlorophyll content of creeping bentgrass, as well as the lowest electrolyte leakage amongst all tested plant growth regulators at 50 mM NaCl.

Both fungal and bacterial endophytes typically colonize and live within an individual host plant throughout its life cycle without causing any significant parasitic symptoms as symbiotic relationships. Among turfgrasses, several species have been identified to mitigate salinity stresses [127]. The mechanisms are highly related to the host plant water and nutrient uptake, and enhanced plant hormone productions to minimize salinity, drought, and heat stresses. Turf quality and carotenoid content were positively correlated with the incidence of the phyla *Chloroflexi* and *Fibrobacteres* in rhizosphere soil, and indole acetic acid level was positively correlated with the phyla *Actinobacteria* and *Chloroflexi* in the roots. The salt-tolerant bacterium Enterobacter ludwigii B30 was isolated from *Paspalum vaginatum*. A salt tolerance test showed that B30 could grow normally in a 500 mM NaCl medium [127].

### 5.5. Symbiotic Relationship with Soil Microorganisms and Genetic Modifications

Responses of *Puccinellia distans* (Table 2), a halophytic grass with low (50 mM) and high (200 mM) NaCl salinity, were studied in a sand culture experiment with and without inoculation by the arbuscular mycorrhizal fungus *Claroideoglomus etunicatum* isolated from its saline habitat [128]. Plant biomass was found to be uninfluenced by salinity levels, but a tendency to develop a higher biomass was observed in arbuscular mycorrhizal fungus plants under the lower and higher saline conditions. Interestingly, leaf photosynthesis increased regardless of salinity or arbuscular mycorrhizal fungus inoculation. Arbuscular mycorrhizal fungus-inoculated plants demonstrated a higher water-use efficiency under the higher saline condition. Arbuscular mycorrhizal fungus inoculation significantly increased leaf osmotic potential. Arbuscular mycorrhizal fungus colonization diminished salt-induced malondialdehyde accumulation, although antioxidative enzymes responded differently. K and Ca contents were not affected by the treatments. Arbuscular mycorrhizal fungus inoculation increased salinity tolerance also by improving water relations and protections against oxidative damage in the leaves [128]. Due to variable environmental factors, the influence of arbuscular mycorrhizal fungus on plant biomass was found to be unrelated to plant phylogeny. The greater biomass accumulation in arbuscular mycorrhizal fungus plants is partially influenced by improved water status, photosynthetic efficiency, and uptake of Ca and K in plants irrespective of salinity stress. In many cases, the uptake of N and P was higher in arbuscular mycorrhizal fungus plants as the salinity increased. Additionally, the activities of malondialdehyde, peroxidase, and superoxide dismutase, as well as the proline content, changed due to arbuscular mycorrhizal fungus inoculation under salinity stress [129].

Genetic diversities (Table 2) exist from a morphologic level to a molecular level among turfgrass germplasm resources [4,130,131,132,133,134], and the extents of variation in their salinity tolerance have been intensively studied. A salt-induced CdWRKY50 was isolated and analyzed in wild bermudagrass [135]. The expression of CdWRKY50 was prominently induced by salt, drought, cold, and abscisic acid (ABA) treatments. Subcellular localization analysis revealed its localization in the nucleus. CdWRKY50-silencing bermudagrass conferred enhanced salt tolerance. In this study, authors also found that transgenic *Arabidopsis* plants overexpressing CdWRKY50 showed decreased salt tolerance. Salt- and antioxidant-related genes, including AtRD29A, AtRD29B, AtDREB2A, AtDREB2B, AtSOS1, AtSOS3, AtSOD1, and AtCAT1 were all clearly repressed in CdWRKY50-overexpressed *Arabidopsis* plants. In conclusion, CdWRKY50 plays a key role in the negative regulation of salt stress within bermudagrass, which provides a new perspective for the underlying molecular mechanism of the CdWRKY50 gene involved in salt stress response [134,135].

## 6. Conclusions

The greatest advantage for improving turfgrass salinity tolerance is the diversity of turfgrass species and cultivars (Table 2). The most significant challenge facing future improvement is the general lack of in-depth research compared to that of common agricultural crops. Two of the main reasons may be that turfgrasses are not food- or fiber-related crops, and there are more severe restrictions on potable water uses for turfgrass irrigation worldwide. Another major challenge is that turfgrass salinity stress is closely associated with other environmental stresses such as drought, heat, soil alkalinity, nutrient imbalances, and other poor soil conditions. Large-scale collaboration with multiple research units seem to promise great potential in facing such challenges. In recent years, soil salinity has become one of the most prominent plant-growth-limiting factors, especially for highly maintained turfgrass landscapes. As many environmental factors, management practices, and soil properties contribute to the cause of salinity stress in plants, the lack of freshwater for irrigation purposes has driven many producers to utilize saline irrigation water sources, thus negatively impacting soil health and fertility. Research over the past few decades has indicated significant metabolic and visual turfgrass responses to soil salinity that negatively affect plant growth and development. Reports have indicated significant differences of salinity tolerance on the varietal level of both cool-season and warm-season turfgrass species. Among these factors, the most important one is that turfgrass salinity tolerance is strongly controlled by a cultivar’s specific genome, as many active and passive resistance mechanisms have been observed in turfgrass species and cultivars when exposed to saline conditions. Thus, understanding the effects of soil salinity on turfgrass’ physiological processes will allow for improvements in future breeding and management practices to adapt to the worldwide issue. This review has documented the current state of knowledge on intra- and inter-species differences in salinity resistance by describing recent research on specific plant responses and resistance mechanisms to soil salinity. Certain management practices that alleviate salinity stress are critical factors in maintaining plant growth and development in saline conditions.

Turfgrass salinity stress has been well documented for decades. However, as the severity and commonality of saline soils have increased in recent years, there is high demand for new technology to aid turfgrass managers in adapting or combatting saline conditions. In efforts to adapt to soil salinity, further salinity tolerance screening should be conducted on warm-season grasses, especially common (*Cynodon dactylon*) and hybrid bermudagrass (*C. dactylon × C. transvaalensis*) species, as they are commonly used in regions that are prone to salinization as the largest single cultivated grass land area (both turfgrass and forage) in the U.S. In addition, future research should include a recovery component after salinity stress to provide more details for consumers regarding a specific turfgrass cultivar’s recuperative capacity to salinity stress. As prior studies have indicated, exogenous applications of various chemical products such as PGRs, silicon, various external applicants (Table 3), and even biological agents have elevated abiotic stress resistance and improved carbohydrate metabolism while under stress for many salt-sensitive turfgrass species [108,123,127]. There is still a need for further research on optimal turfgrass nutrition and nutrition homeostasis before and during salinity stress. Overall improvement in turfgrass salinity resistance requires focused breeding efforts as well as extensive screening protocols to develop new cultivars that can grow in saline conditions while maintaining essential management and playability characteristics as three important aspects for the future.

## Figures and Tables

**Figure 1 plants-12-00925-f001:**
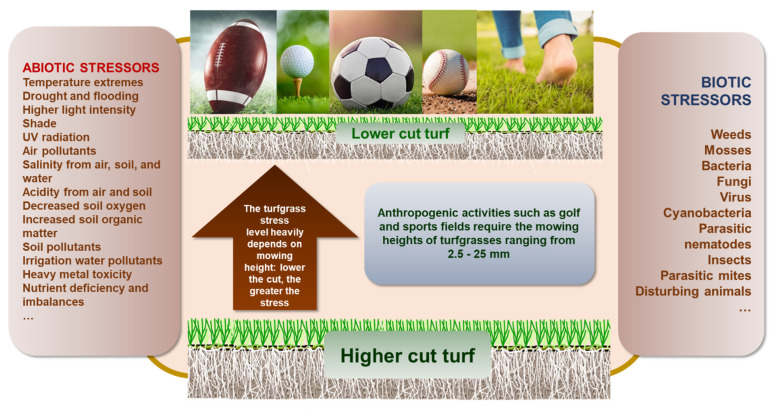
Turfgrasses encounter both abiotic and biotic stresses with mowing height as one of the most critical factors affecting the severity of stresses. A maintained turf area may receive more than one stress at a time or season, and it is very common that a turfgrass must be resistant to multiple stressors in addition to the lower mowing height stress for functional and performance purposes.

**Figure 2 plants-12-00925-f002:**
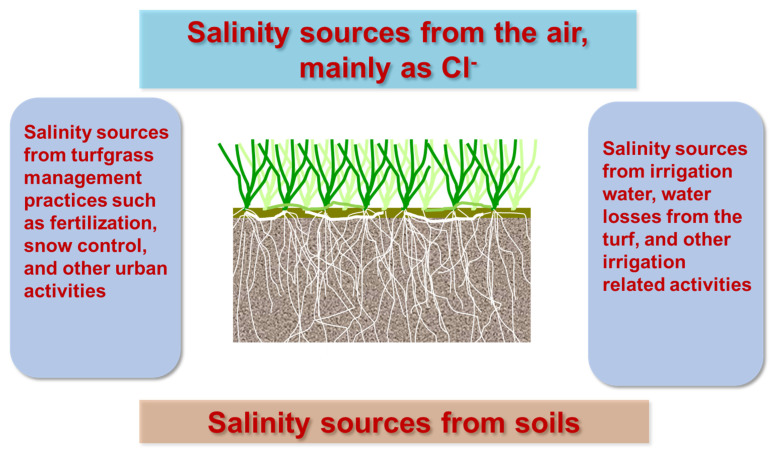
Multiple sources contribute to the salinity stresses of a turfgrass community, and these sources vary depending on location conditions.

**Figure 3 plants-12-00925-f003:**
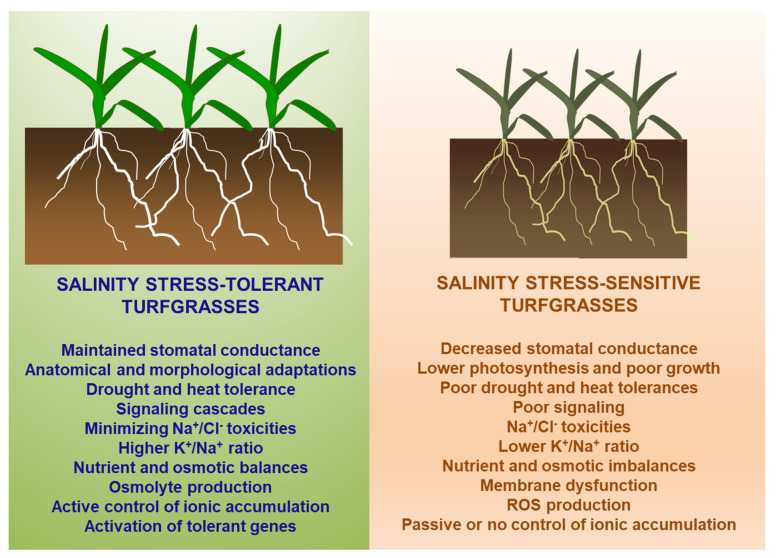
Metabolic processes associated with salinity tolerances and sensitivities among turfgrass species and cultivars.

**Table 1 plants-12-00925-t001:** Classification of salt-affected soils.

Class	EC	ESP	SAR	Soil pH
	dSm^−1^	%		
Saline	>4.0	<15	<12	<8.5
Sodic	<4.0	>15	>12	>8.5
Saline-Sodic	>4.0	>15	>12	<8.5

EC: Electrical Conductivity. ESP: Exchangeable Sodium Percentage. SAR: Sodium Absorption Ratio.

**Table 2 plants-12-00925-t002:** Relative salinity tolerance among turfgrass species and their hybrids with estimated cultivar variations within each species (within the same category of the same background color, the order is following the scientific name in alphabetical order).

Intolerant 0–2 dS m^−1^ 0–22 mM NaCl	Slightly Tolerant 2–4 dS m^−1^ 22–44 mM NaCl	Moderately Tolerant 4–8 dS m^−1^ 44–110 mM NaCl	Tolerant 8–30 dS m^−1^ 110–410 mM NaCl	Halophytes >30 dS m^−1^ >410 mM NaCl
**Average ocean water salinity level = 35,000 ppm; 44 dS m^−1^; or equivalent to 600 mM NaCl**				

			***Distichlis spicata*; C_4_ Inland saltgrass**	
			***Paspalum vaginatum*; C_4_ Seashore paspalum**	
			***Puccinellia* spp.; C_3_ Alkaligrasses**	
			***Sporobolus virginicus*; C_4_ Seashore dropseed**	
			***Zoysia macrostachya*; C_4_ Macrospike zoysiagrass**	

			***Zoysia japonica;* C_4_ Japanese lawn grass**	
			***Zoysia matrella;* C_4_ Manila zoysiagrass**	
			** *Zoysia pauciflora* **	
			***Zoysia sinica;* C_4_ Chinese zoysiagrass**	
			**Zoysia spp.; C_4_ Hybrid zoysia grasses**	

		***Buchloe dactyloides*; C_4_ Buffalograss**		
		***Cynodon dactylon*; C_4_ Common bermudagrass**		
		***Cynodon transvaalensis*; C_4_ African bermudagrass**		
		***Zoysia tenuifolia;* C_4_ Mascarene grass**		
		***Zoysia* spp.; C_4_ Hybrid zoysiagrasses**		


	* **Agrostis alba** * **; C**_**3**_ **Redtop bentgrass** ***Agrostis canina*****; C**_**3**_ **Velvet bentgrass** ***Agrostis capillaris*****; C**_**3**_ **Colonial bentgrass** ***Agrostis***** spp.; C**_**3**_ **Hybrid bentgrasses** ***Agrostis stolonifera*****; C**_**3**_ **Creeping bentgrass** ***Axonopus***** spp.; C**_**4**_ **Carpetgrasses** ***Bouteloua gracilis*****; C**_**4**_ **Blue grama grass** ***Eremochloa ophiuroides*****; C**_**4**_ **Centipedegrass** ***Fescue ovina*** **var.** ***ovina*****; C**_**3**_ **Sheeps fescue** ***Festuca arundinacea*****; C**_**3**_ **Tall fescue** ***Festuca longifolia*****; C**_**3**_ **Hard fescue** ***Festuca rubra*** **subsp.** ***commutata*****; C**_**3**_ **Chewings fescue** ***Festuca rubra*** **var.** ***rubra*****; C**_**3**_ **Creeping red fescue** ***Koeleria macrantha*****; C**_**3**_ **prairie Junegrass** ***Lolium perenne*****; C**_**3**_ **Perennial ryegrass** ***Lolium*** **spp.; C**_**3**_ **Hybrid ryegrasses** ***Pennisetum clandestinum*****; C**_**4**_ **Kikuyugrass** ***Poa pratensis*****; C**_**3**_ **Kentucky bluegrass** ***Poa*** **spp.; C**_**3–4**_ **Hybrid bluegrasses** ***Poa supina*****; C**_**3**_ **Supina bluegrass** ***Poa trivialis*****; C**_**3**_ **Rough-stock bluegrass** ***Stenotaphrum secundatum*****; C**_**4**_ **St. Augustinegrass**			

	** * Paspalum notatum * ; C_4_ Bahiagrass **			
	** * Lolium multiflorum; * C_3_ Annual ryegrass **			
	** * Poa * spp.; C_3_ Hybrid bluegrasses **			

** * Poa annua * ; C_3_ Annual bluegrass **				

## Data Availability

Not applicable.

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
