# Peer review of "Turfgrass Salinity Stress and Tolerance—A Review"

_plants, 2023, doi:10.3390/plants12040925_

Round 1

Reviewer 1 Report

This is a useful and comprehensive review of salt tolerance/sensitivity in wide range of economically important turfgrasses.

The article is long and to be reader-friendly, it needs to be better organized, using more subheadings to describe different types of stress and the various plant defences against these.   

Lines 66, 67: “Sodic soils most commonly get interchanged with saline soils, the difference between the two is that sodic soils have high concentrations of sodium (Na+) and chloride (Cl-) salts (ionic soluble salts).” The second “sodic” should be “saline”, right?

The ability of plants to survive in saline media/environments is still under intensive research. Some progress was made recently in plant ancestors, the Characeae. In Characeae, Na+ enters the cells through nonselective cation channels. The permeability of these channels is decreased by presence of Ca2+. Thus higher Ca2+ in the soil/medium improves salt tolerance. Salt tolerant Chara longifolia responds to saline stress by activating Na+ ATPase (Phipps, Delwiche and Bisson, 2021, J. Phycol. 57: 1004 -1013; Phipps, Delwiche and Bisson, 2021, J. Phycol. 57: 1014 -1025). Salt-sensitive Charas seem to lack this transporter. All Characeae contain H+ ATPases and Na+/H+ antiporters. It might be interesting to examine the available turfgrass genomes for molecular fingerprint of Na+ ATPase.

Author Response

Dear Reviewer 1:

Many thanks for your time in reviewing this manuscript and most your suggestions have been implemented in the manuscript. Based on your suggestion, we added subtitles for Sections 4 and 5.

LIne 66-67 "sodic" was correct here.

The suggestion to add new findings on Ca2+ is a very good one adding to our knowledge, but with the consideration to limit the length of the manuscript, we appreciate the update on the subject to focus on turfgrasses. Calcium has been always one of our favorite topics and we may develop new research in the future as you suggested.

Reviewer 2 Report

The review on Turfgrass Salinity Stress and Tolerance is useful. I may have the following major and minor comments.

1.     Can you make your title more specific

2.     The key conclusion from your abstract is not clear. You need to be very sharp on your key conclusions. The conclusions and implications are not much attractive. The knowledge gaps are not clear.

3.      It is unclear from your introduction that why do you want to write this review? How will you review advance the understanding? Why is your review important and timely?

4.     There is quite large room to improve the writing. Not only for the writing itself but also for the writing logic. The research motivation are not clear. 

5.      Several unnecessary abbreviations are preventing the reading. I have to remember a lot abbreviations when reading your manuscript.

6.     Some sentences are unnecessary long with changing focuses. It is not easy to understand these long sentences. The writing needs to be improved.

7.     The main conclusions and key implications are not clear enough from your discussion. The potential mechanisms need to be discussed. What are the key implications? what are the major knowledge gaps and future research priorities?

8.     Regarding your references, 1. Be consistent with either upper- or lower-case letters in the title. 2. Pay attention to the author names.

Author Response

Many thanks and please see the attached pdf file.

Reviewer 3 Report

Dear Authors,

The review article "Turfgrass Salinity Stress and Tolerance" documents the current state of knowledge of the effects of salt stress on turfgrasses as well as the resistance, tolerance, and avoidance mechanisms of individual species to salt stress.  

The title of the manuscript seems appropriate.

Abstract: The abstract is representative and informative.

Introduction: The first paragraph is very similar to the abstract. I suggest some changes to avoid unnecessary repetition.

Conclusions: well written. 

Literature: Appropriate and up-to-date literature is cited.

There are some minor corrections in the PDF file.

Author Response

Dear Reviewer 3,

Thank you very much for your time in reviewing this manuscript and those changes were made based on your suggestions.

Sincerely,

The manuscript authors

Reviewer 4 Report

The draft is a nice write-up, but I do not see any substantial value added by it. Further, the manuscript does not provide any new perspective or interpretation to the broader audience either. I regret to say that I do not feel it could be a good candidate for Plants.

Author Response

Dear Reviewer 4,

We thank you very much for your time in reviewing the manuscript.

Sincerely,

The manuscript authors